# Lives saved and lost in the first six month of the US COVID-19 pandemic: A retrospective cost-benefit analysis

Olga Yakusheva[1,2]*, Eline van den Broek-Altenburg[3], Gayle Brekke[4], Adam Atherly[3]

1 Department of Systems, Populations and Leadership, School of Nursing, University of Michigan, Ann Arbor, MI, United States of America, 2 Department of Health Management and Policy, School of Public Health, University of Michigan, Ann Arbor, MI, United States of America, 3 Larner College of Medicine, University of Vermont, Burlington, VT, United States of America, 4 University of Kansas Medical Center, Kansas City, KS, United States of America

* yakush@med.umich.edu

**Data Availability Statement:** All relevant data are within the manuscript. All data required to replicate the findings in the study are included in the text of

## Abstract

In the beginning of the COVID-19 US epidemic in March 2020, sweeping lockdowns and other aggressive measures were put in place and retained in many states until end of August of 2020; the ensuing economic downturn has led many to question the wisdom of the early COVID-19 policy measures in the US. This study's objective was to evaluate the cost and benefit of the US COVID-19-mitigating policy intervention during the first six month of the pandemic in terms of COVID-19 mortality potentially averted, versus mortality potentially attributable to the economic downturn. We conducted a synthesis-based retrospective cost-benefit analysis of the full complex of US federal, state, and local COVID-19-mitigating measures, including lockdowns and all other COVID-19-mitigating measures, against the counterfactual scenario involving no public health intervention. We derived parameter estimates from a rapid review and synthesis of recent epidemiologic studies and economic literature on regulation-attributable mortality. According to our estimates, the policy intervention saved 866,350–1,711,150 lives (4,886,214–9,650,886 quality-adjusted life-years), while mortality attributable to the economic downturn was 57,922–245,055 lives (2,093,811–8,858,444 life-years). We conclude that the number of lives saved by the spring-summer lockdowns and other COVID-19-mitigation was greater than the number of lives potentially lost due to the economic downturn. However, the net impact on quality-adjusted life expectancy is ambiguous.

## Introduction

In response to the COVID-19 pandemic in the US, a series of lockdowns mitigating the spread of the pandemic were introduced beginning in March and maintained into July of 2020 when the first wave of the pandemic was largely believed to have passed. With millions of lives and trillions of dollars at stake, the wisdom of imposing lockdowns to control a pandemic is controversial in both the academic literature and the lay press [1–9]. Much of the debate revolves

the manuscript, all explanations for replicating the results (formulas, step-by-step instructions) are also provided in the text. Citations to peer-reviewed academic papers from which the data were derived are provided in the text.

**Funding:** The author(s) received no specific funding for this work.

**Competing interests:** The authors have declared that no competing interests exist.

around the tradeoff between lives saved and economic impact, a question routinely addressed through economic evaluations.

Economic evaluations of health-related policies typically compare policy-attributable health benefits and costs in monetary terms by assigning a dollar value to the lives saved or to the resulting life expectancy gained. One problem with this approach is that economic estimates of the value of a human life vary between $8-$11 million per life. Several cost-benefit calculations of the COVID-19 lockdowns estimated the monetary value of lives saved from as low as $1.15 trillion to as high as $65 trillion [7, 9–14], thus only intensifying the national debate about whether the lockdowns were worth the cost. But beyond the challenge of calculating the correct value for cost-benefit calculations, the economic approach of assigning average monetary values to a human life frequently fails to resonate with the public, including individuals worried about themselves and loved ones and with healthcare professionals who took an oath to save lives [3, 15–17].

This study offers an innovative approach to the dollars-per-life-saved conundrum by instead estimating the number of non-COVID-19 deaths potentially attributable to the economic downturn brought on by the lockdowns and other COVID-mitigating measures (regulation-attributable mortality). The COVID-19 lockdowns were not the first time that public safety and wellbeing required restrictions on personal freedoms and economic activity. For example, road safety regulations (e.g. seatbelt laws, lane width and marking codes, and speed limits) save lives but at the same time they are costly to implement and enforce [18]. Many government programs (e.g., occupational safety regulations, homeland security programs) consume taxpayer dollars, therefore diverting them from personal spending on food, housing, and healthcare [19–21]. Since a loss of income is known to negatively impact individuals' health and wellbeing [22–32], these life-saving regulations, depending on their cost, may end up being counter-productive. For example, a study of regulation-attributable mortality of fire-mitigating programs in Australia found that, at an annual taxpayer cost of $12 billion (in 2020 US Dollars), fire regulations in Australia potentially cause as many deaths from a reduction in people's incomes as the number of fire fatalities they prevent [33]. Importantly, these analyses of mortality attributable to government regulations do not use the value of statistical life approach—instead, they empirically derive the economic cost that would induce one statistical death based on empirical relationships between income and mortality observed in large national cohorts.

Much like other government safety regulations, COVID-19 lockdowns and other measures were put in place to protect lives, but they also led to a loss of personal income for many. Therefore, the aim of this study was to compare lives saved by the COVID-19-mitigating policy intervention during the spring and summer of 2020 to regulation-attributable mortality potentially caused by the ensuing economic downturn in the US. Our hope is to initiate a broader discussion whether a 'lives-to-lives' comparison could allow for assessments of the judiciousness of economic lockdowns without the distraction of normative assessment differences in the value of human life.

## Materials and methods

This study is a synthesis-based retrospective cost-benefit evaluation of the US COVID-19-mitigating policy intervention in the US population of all ages during the first six months of the pandemic. Taking a societal perspective, we considered the "benefit" to be COVID-19 mortality potentially averted by the intervention. The "cost" was the economic downturn (loss of national income during the first 6 months of the pandemic) and its attributable mortality.

Parameter estimates for our calculations were obtained via rapid review [34, 35] of academic literature and reports by official US government and international bodies.

## Intervention

This study evaluated the COVID-19-mitigating public policy intervention as the entire complex of federal, state, and local COVID-19-mitigating measures, including lockdowns and all other measures implemented during the first 6 months of the pandemic (March through August 2020) in the US. We define lockdowns as government-induced mandatory restrictions on private activity, including closures of businesses and public gatherings (e.g., schools, public offices), building capacity regulations, and stay-at-home orders. Other COVID-19-mitigating measures included all non-mandatory public health measures including mask-wearing and hand-washing guidelines, recommendations to refraining from family gatherings, and social distancing. The comparative strategy was 'no intervention,' a hypothetical scenario where no COVID-19-mitigating restrictions or measures were implemented in the US during the first six months.

## Health outcomes

The primary outcome for evaluating cost and benefit of the COVID-19-public health intervention was cumulative mortality, measured as the number of lives (gained and lost). Our secondary derivative outcome was cumulative life expectancy measured as the number of quality-adjusted life-years (gained and lost).

## Time horizon

The time horizon was March 1 through August 31 2020, chosen because the most severe restrictions (sweeping lockdowns, shut-down of non-essential sectors of the economy) were in place during the first six months of the pandemic, and also because epidemiological models of the unmitigated US epidemic predicted it to end by September of 2020 without any mitigating interventions, thus allowing us to attribute the full potential US mortality from COVID-19 under the counterfactual no-intervention scenario to the March 1 through August 31 period [36–39].

## Analysis

The number of lives saved by the COVID-19-mitigating policy intervention during the first six months of the US pandemic was calculated in two steps. First, to calculate the potential unmitigated cumulative COVID-19 mortality, we multiplied the US population, 330 million, by the COVID-19 herd immunity threshold (HIT) and by the infection fatality rate (IFR). Then, to calculate the number of lives potentially saved by the intervention, we subtracted out the observed cumulative COVID-19 mortality reported during the same period. Life expectancy gained was calculated as the number of lives potentially saved by the intervention times 5.64 years—the quality-adjusted life expectancy lost among COVID-19 fatalities [40].

Assuming cost-to-death ratio (CDR) estimates and methods from the economic literature on regulation-attributable mortality [17–21, 33, 41, 42] are applicable to evaluating mortality attributable to the COVID-19 policy intervention, the number of lives lost from the economic downturn was calculated by dividing the combined first and second quarter US GDP loss, $2.23 trillion [43], by the CDR. The CDR measures the cost of a government intervention that induces one statistical death. Dividing the full economic cost of a regulation by the CDR yields the total number of regulation-attributable deaths. Quality-adjusted life expectancy lost was

calculated as the number of lives lost due to the economic downturn times 36.1 quality-adjusted life years per life lost—the 2019 average US life expectancy (78.7 years) minus the 2019 median age (37.9 years) [44, 45] multiplied by the health-related quality of life score (0.886) [46].

Due to the short-term timespan of our analysis, we did not use discounting.

## IFR, HIT, CDR parameters

To obtain the infection fatality ratio (IFR) and the herd immunity threshold (HIT) parameters, we reviewed six projections of COVID-19 cumulative mortality by US and international authorities [37, 38, 47–49], a systematic review and meta-analysis study of the SARS-CoV-2 IFR [50–52], and four recent studies of HIT in the US [53–56]. To obtain the CDR, we searched EconLit, the official reference search tool of the American Economic Association. We used a free-text search using terms "cost-per/to-death," "regulation-attributable mortality/fatalities," "mortality/fatality cost," and "government regulation/intervention." We included studies that: 1) were published after 1990; 2) used data from the US, Europe, or Australia; 3) empirically estimated the CDR from large, representative cohorts or national databases; and 4) reported adjusted (not only crude) CDR estimates. Studies that used the value of a statistical life instead of the CDR were excluded [17, 21, 42]. Reference lists of the included studies were searched for additional studies meeting the above criteria. We synthesized the evidence based on the analysis method, study population, and type of covariate adjustment. If a study reported several analysis approaches, we extracted the most rigorous CDR estimate using the following criteria: 1) adjusted cross-sectional estimates were preferred to unadjusted (crude) estimates, 2) longitudinal (cohort, panel, time-series) estimates were preferred to cross-sectional estimates, and 3) individual-level analyses were preferred to ecological studies. We converted all CDRs to current 2020 US dollars by using historical data on the Consumer Price Index from the US Bureau of Labor Statistics.

## Results

### Lives saved

Given the evolving stage of knowledge about COVID-19's IFR and HIT, we included a broad range of IFR and HIT estimates from the literature and calculated a low and high bounds for the number of lives saved by the intervention. For IFR, we attempted to use estimates obtained from data during the first six months of the pandemic when less was known about the disease and few effective treatment approaches existed. A median IFR of 0.23 was reported by the WHO [52] across 51 locations globally; among the 11 US states included in the report, the IFR ranged from 0.08 in Utah and 0.20 in Missouri, to 1.54 in Connecticut and 1.63 in parts of Louisiana. Overall, locations that had higher number of deaths (most of the US) had a median IFR of 0.57. A recent meta-analysis of 24 IFR studies reported the IFR of 0.68 with the 95% confidence interval between 0.53% and 0.82%, subsequently supported in an age-specific meta-analysis of 111 studies from 33 locations. The HIT is thought to be between 60–70% [53–56].

Applying 0.53%-0.82% IFR and 0.60–0.70 HIT to the US population of 330 million, the unmitigated mortality (in the absence of any intervention) was 1,049,400–1,894,200 deaths. After subtracting the 183,050 confirmed COVID-19 deaths on August 31 2020 [57], the number of COVID-19 lives saved by the first six months of the policy intervention was 866,350–1,711,150 lives. Measured in cumulative quality-adjusted life expectancy instead of number of lives saved, life expectancy gained was 4,886,214–9,650,886 quality-adjusted life-years.

## Regulation attributable mortality

Seven economic studies of mortality attributable to government regulation satisfied our inclusion and exclusion criteria (**Table 1**) [18–20, 22, 23, 33, 41]. Among them, four derived the

**Table 1. Evidence table and regulation-attributable mortality calculations.**

| Study | | | Cost-to-death (CDR) estimate[a] | | | Regulation-attributable mortality[a,b] | |
|---|---|---|---|---|---|---|---|
| | | | As reported in the study | Inflation adjustment | In current 2020 USD | Lives lost | Comorbidity-adjusted life-years lost |
| Ashe, B., de Oliveira, F. D., & McAneney, J. (2012) | Fire management to prevent structural and bushfires in Australia | Cross-sectional estimates from (Keeney 1997), age and sex adjusted, rescaled to Australian income and mortality data from the 2006 Australian Census data, Australian adults age 35 and older, 2006 | $20; $50 mil 2010 AU | 0.77[c] | $15.4; $38.5 mil | 144,805; 57,922 | 5,908,052; 2,363,221 |
| Chapman, K. S., & Hariharan, G. (1994) | Select health and safety regulations during 1970–1990 | Cross-sectional survival analysis of 10-year mortality and personal income; adjusted for age, wealth, employment, family structure, health and disability; 1969–1979 Retirement History Survey merged with Social Security records through 1974, US males 58–62 in 1969 | $12.2 mil[d] 1990 US | 1.96 | $23.9 mil | 93,305 | 3,806,862 |
| Chapman, K. S., & Hariharan, G. (1996) | Occupational safety regulations | Panel survival analysis of annual mortality and personal income; adjusted for age, wealth, employment, education, family structure, number of living parents, and health status; 1966–1990 National Longitudinal Survey of Mature Men, older US men 45–59 in 1966 | $6.4; $8.7 mil 1990 US | 1.96 | $12.5; $17.0 mil | 178,400; 131,176 | 7,278,720; 5,352,000 |
| Gerdtham, U.-G., & Johannesson, M. (2002) | Unspecified government regulation | Panel survival analysis of annual mortality risk and personal disposable income; adjusted for sex, age, wealth, income, disposable income, education, employment, immigrant status, family structure, health status, blood pressure, functional limitations; Swedish adults 20–84, 1980–1986 | $6.8; $9.8 mil 1996 US | 1.63 | $10.8; $16.0 mil | 206,481; 139,375 | 8,424,445; 5,686,500 |
| Elvik, R. (1999) | Road safety regulations | Time series analysis of mortality and income per capita during ten 5-year periods between 1946 and 1995; adjusted for age and sex, data for from Statistics Norway 1996–97 | $7.1 mil[e] 1995 USD | 1.68 | $11.9 mil | 187,395 | 7,645,714 |
| Keeney, R. L. (1990) | Unspecified government regulation | Cross-sectional annual mortality rates across income groups; adjusted for age and sex; 1970 data, white US adults 25–64 | $3.14; $7.25 mil 1980 US | 3.11 | $9.8; $22.5 mil | 227,551; 99,111 | 9,284,082; 4,043,733 |
| Keeney, R. L. (1997) | Unspecified government regulation | Cross-sectional annual mortality risk across income groups, age and sex adjusted, using National Longitudinal Mortality Study 1979–1985 data, 550,000 US adults 25–64 | $4.85; $13.33 mil 1991 US | 1.88 | $9.1; $25.1 mil | 245,055; 88,845 | 9,998,242; 3,624,864 |

Notes

[a] Estimates are reported as the estimate under the proportional cost allocation assumption followed by the estimate under the progressive cost allocation assumption, separated by a semicolon.

[b] Author calculations for $2.23 trillion GDP loss.

[c] Costs reported in AU$2010, the 0.77 multiplier adjusts for 2020 AU$ to US$ exchange rate 0.65, and US 2010–20 inflation coefficient 1.88.

[d] Progressive cost allocation is assumed.

[e] Proportional cost allocation is assumed.

CDR from cross-sectional differences in annual mortality rates across income groups [18–20, 33], two estimated an individual-level panel regression of annual mortality on personal income [22, 23], and one estimated a time-series regression of annual mortality rate on per-capita national income over several decades [18]. Four of the studies used US data [19, 20, 22, 23], two used Northern European data [18, 41], and one used Australian data [33]. All of the studies adjusted for basic demographic factors (sex, age) and three adjusted for an extensive set of covariates, including family characteristics (e.g. marital status, children) and health (e.g. self-reported health status, disability or functional limitations) [22, 23, 41]. Five studies examined how the CDR may depend on the allocation of the cost burden of a regulation across income groups—proportionally to income (flat rate) or progressively with income (higher income groups affected relatively more than lower income groups) [19, 20, 22, 33, 41], while one study assumed proportional cost allocation [18] and one assumed progressive cost allocation [23]. The CDR ranged $9.1—$15.4 million (in 2020 Dollars) under the proportional cost allocation assumption and $16—$38.5 million under the progressive cost allocation assumption. The two most rigorous studies utilizing a panel-data analysis with adjustments for an extensive set of controls [22, 41] reported the CDR from $10.8 to $12.5 million and $16 to $17 million under the proportional and progressive cost allocations, respectively.

Regulation-attributable annual mortality from the $2.23 trillion economic loss ranged from 144,805–245,055 deaths and 57,922–139,375 deaths under the proportional and the progressive cost allocation assumptions, respectively. The attributable mortality was lower (and the CDR was higher) under progressive cost allocation because higher income individuals are less vulnerable to income reductions. [22, 24, 31]. Taking the minimum and the maximum of these numbers, we estimated regulation-attributable mortality to be 57,922–245,055 deaths. Measured in cumulative life expectancy instead of number of lives, life expectancy lost was 2,093,811–8,858,444 life-years.

## Discussion

This study is the first to evaluate the societal costs and benefits of the US public health response during the first six months of the COVID-19 pandemic not in monetary terms but in terms of human lives saved and potentially lost. Our calculations suggest that the number of lives potentially saved by the spring 2020 lockdowns and other mitigating measures impacting the US economy (866,350–1,711,150 lives) far exceeds the number of lives potentially lost during the same time period due to the ensuing $2.23 trillion economic downturn (57,922–245,055 lives). However, because the majority of lives saved are those of older adults with multiple chronic illnesses whose life expectancy is shorter on average, the impact of the intervention on cumulative life expectancy is less clear (4,886,214–9,650,886 quality adjusted live-years saved; 2,093,811–8,858,444 quality-adjusted life-years lost). From an ethical perspective, a potential caveat to the quality-adjusted life-years approach is that, by design, it assigns a smaller value to the lives of older adults compared to younger adults, and persons with disabilities compared to fully-abled persons [58, 59]; racial and ethnic disparities in life expectancy have also been well-documented [60]. To avoid exercising judgement and bias about the value of a human life, the calculation based on the number of lives saved and lost might be the preferred approach.

Lockdowns and other restrictions on private activity have real humanitarian consequences that should not be overlooked. Known as the income gradient in health, the notion that a person's health and life expectancy is in large part determined by their income is not new [22–32]. The association of income with life expectancy is the strongest among lowest-income individuals who have minimal discretionary spending to cushion the blow [22, 24]. Poverty can limit access to basic needs like transportation, health care, shelter, and clean food and water [22, 24,

31]. Even transient income fluctuations have been linked to reduced essential household expenditures (e.g. food and shelter) and loss of access to health insurance, and to negative health impacts including lower self-reported health [61–65], increased risk of cardiovascular disease, reduced brain health, and increased all-cause mortality [26, 27, 66]. A large amount of literature across disciplines evaluated health and mortality effects of the most recent global Great Recession of 2007–2009 which caused a similar magnitude of economic decline to the current COVID-19 recession [67]. The preponderance of the evidence is consistent with negative health effects on fertility, self-rated health, overall morbidity (including both new-onset and exacerbation of existing chronic illness) and increased mortality for older adults, racial and ethnic minority groups, and individuals in countries with weaker social support structures [67].

It is important to note that the 2020 economic downturn disproportionally impacted the poorest, most marginalized members of our society. According to experts, this recession could be the most unequal in modern U.S. history, as job losses are overwhelmingly affecting minority workers, younger workers, and the less-educated working-poor [68, 69]. Three months after the spring-summer lockdowns had been largely lifted, only one out of three Black Americans who lost their job during the lockdowns regained employment [68]. The disproportional impact of the recession on low-income groups suggests that the indirect mortality attributable to the lockdowns and other COVID-19-mitigating measures may be closer to the more pessimistic projections and higher end of our lives lost estimates, close to 250,000 deaths and 9 million years of lost human lifespan. It is also important to note that while the costs of the lockdowns disproportionally fell on the younger, less educated, low-income workers, the benefits are disproportionally accruing to older adults who are at the highest risk of dying from COVID-19. A better understanding of these distributional effects is needed to inform current and future policy how to anticipate and better manage public sentiment during public health crises requiring a strong government response.

Our calculations are based on the best available evidence at this time. To calculate lives and quality-adjusted life expectancy saved by the intervention, we used IRFs between 0.53% and 0.82% from systematic reviews of data collected during the first 6 months of the pandemic [50, 51]. These estimates are based on observed mortality and serological prevalence data, and therefore incorporate epidemiological and healthcare trends in COVID-19 mortality during that time (e.g., virus becoming less deadly, doctors getting more experience in treating COVID-19, medical supplies becoming more readily available). Our IFR range is considerably lower than the March 2020 estimates based on early data from the Wuhan province in China (IFR between 1.1% and 3.4%) which at the time gave rise to early predictions of potential unmitigated death toll between 2.2 and 2.9 million lives in the US [36–39]. Therefore, our estimate of COVID-19 lives saved, 0.87–1.7 million, reflects our current best understanding of SARS-CoV-2 and potential population-level mortality from COVID-19.

Our results are subject to several qualifications. First, we conducted an "as-is" ex-post evaluation of the public health intervention, including the lockdowns and other COVID-19 mitigating measures (including any voluntary behavioral modification component). This is only a first step, and future research of incremental effectiveness of different public health measures (lockdowns, school closures, travel restrictions) is needed to determine whether a more targeted, nimble, or shorter set of measures may have resulted in a more favorable net balance of lives saved versus lost. Additionally, we only evaluated the initial six months of the US pandemic; therefore, our findings have limited ability to inform decisions regarding any future use of lockdowns and other measures. Specifically, our estimates are based on infection fatality rates during the early stages of the pandemic when less was known about effective treatments and many healthcare facilities experienced shortages of medical supplies (ventilators, personal

protective equipment). With effective treatment modalities and vaccines now available, the vulnerable can be protected through targeted measures limiting a need for future blanket restrictions.

We attributed all COVID-19 lives saved (relative to the unmitigated counterfactual) to the public health measures (lockdowns, social distancing recommendations, masking recommendations), even though some voluntary behavioral modifications (e.g., limiting social contacts, trips to the store, or non-essential travel outside the state) would likely have taken place among the public even in the absence of these government interventions. It is not possible to empirically isolate self-induced voluntary behavioral changes that the public would have initiated without any government policies from behavioral changes induced by government regulations through either mandates or public awareness campaigns. Inability to quantify the self-initiated (not induced by government intervention) component of behavioral modification means that our analysis may overstate the impact of the government intervention on COVID-19 lives saved, and, likewise, on the economy. However, because behavioral adaptations (whether they are government induced or not) reduce both sides of the equation (benefits and costs), it is unclear whether and how this limitation may have affected our findings on the net.

Although our study is based on best available evidence, the evidence has several important limitations. First, our COVID-19 mortality estimates are based on IFRs that were observed under a full set of public health measures during the first 6 months of the pandemic; therefore, our analysis likely understated the potential cumulative death toll under the hypothetical unmitigated scenario leading us to understating the number of lives and life expectancy saved by the measures. Second, long-term population-level morbidity and mortality among COVID-19 survivors is yet unknown. Emerging sequelae of a COVID-19 infection include damage to the lungs, heart, and brain, suggesting increased risk of long-term health problems and mortality [70, 71]. Not accounting for these long-term effects further increases the possibility that our study underestimated the cumulative mortality potentially averted by the spring 2020 intervention. Third, a salient limitation of the regulation-attributable mortality literature is that cost-to-death estimates are derived from adjusted cross-sectional comparisons of mortality across individuals with different observed incomes, not clearly distinguishing between long-term income differences and short-term income variability. Theoretically, a short term loss of income, like during the economic recession that followed the COVID-19 lockdowns in the US, may impact individuals less (or differently), compared to a persistent economic disadvantage over the course of a lifetime [72, 73]. Further, the negative impact of the economic downturn on personal incomes was partly ameliorated by the US government' COVID-19 economic relief spending, providing assistance to workers, families, and businesses. Therefore, our study may have overstated mortality and loss of life expectancy potentially attributable to government regulation, while possibly understating lives saved and life expectancy gained. Lastly, rapid review and synthesis has inherent limitations relative to systematic reviews [34, 35].

## Conclusion

Evaluated as a full complex of COVID-19-mitigating restrictions, the number of lives saved by the spring-summer lockdowns and other COVID-19 mitigation was greater than the number of lives potentially lost due to the economic downturn, while the net impact on quality-adjusted life expectancy is less clear owing to the older age and poorer health of individuals at highest risk of mortality from the disease. Moving forward, it is essential that we emerge from the pandemic with the smallest humanitarian cost from the combination of disease impact from the virus and the lost economic opportunity.

## Author Contributions

**Conceptualization:** Olga Yakusheva, Eline van den Broek-Altenburg, Gayle Brekke, Adam Atherly.

**Formal analysis:** Olga Yakusheva.

**Investigation:** Olga Yakusheva.

**Methodology:** Olga Yakusheva.

**Writing – original draft:** Olga Yakusheva.

**Writing – review & editing:** Olga Yakusheva, Eline van den Broek-Altenburg, Gayle Brekke, Adam Atherly.

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
