## [Decision Letter · Decision Letter 0]

14 May 2021

PONE-D-21-09447

Lives Saved and Lost in the First Six Month of the US COVID-19 Pandemic: A Retrospective Cost-Benefit Analysis

PLOS ONE

Dear Dr. Olga Yakusheva,

Thank you for submitting your manuscript to PLOS ONE. After careful consideration, we feel that it has merit but does not fully meet PLOS ONE’s publication criteria as it currently stands. Therefore, we invite you to submit a revised version of the manuscript that addresses the points raised during the review process.

We look forward to receiving your revised manuscript.

Kind regards,

Carlos Alberto Zúniga-González, Ph.D

Academic Editor

PLOS ONE

Additional Editor Comments:

Dear, I consider that you have made a good effort to present this manuscript, but it is very important to make the improvements indicated by the reviewers. I believe that if you include and apply the concept of opportunity cost, it could help you explain the cost-benefit relationship, given the benefit-cost relationship does not clarify the cost as we know it regularly in economics or the reader is used to it, so I suggest you include the cost opportunity

Journal Requirements:

2. In the Methods section of the manuscript please provide additional details regarding the rational for the selection of literature used for the selection of IFR and Hit parameters. Furthermore, please clearly indicate the date range for the literature search conducted. Finally, please provide a justification for the selection of data used from only US, Europe and Australia, as an inclusion criteria during the literature search.

Reviewers' comments:

Reviewer's Responses to Questions

**Comments to the Author**

1. Is the manuscript technically sound, and do the data support the conclusions?

Reviewer #1: No

Reviewer #2: Partly

2. Has the statistical analysis been performed appropriately and rigorously? 

Reviewer #1: No

Reviewer #2: N/A

3. Have the authors made all data underlying the findings in their manuscript fully available?

Reviewer #1: Yes

Reviewer #2: Yes

4. Is the manuscript presented in an intelligible fashion and written in standard English?

Reviewer #1: Yes

Reviewer #2: No

5. Review Comments to the Author

Reviewer #1: Report for “Lives Saved and Lost in the First Six Month of the US COVID-19 Pandemic: A Retrospective Cost-Benefit Analysis”

This article evaluates the benefits and costs of U.S. COVID-19 mitigation policy interventions. In terms of benefits, the authors calculate the number of lives saved by subtracting the observed actual number of deaths from the unmitigated potential number of deaths. In terms of costs, the authors estimate the potential number of deaths due to the economic downturn caused by the intervention policies, which is calculated by multiplying the GDP loss in this period with the cost-to-death ratio taken from economic literature.

I have major concerns with respect to the methods used to estimate both the benefits and costs of the intervention policies.

Benefits:

The authors estimate the benefits of intervention policies by subtracting the actual number of deaths from the unmitigated potential number of deaths. By doing so, the authors assume that the decrease in COVID mortality in this time period is all caused by government mitigation policies, which is a questionable assumption in my opinion. First, the virus itself could become less deadly over time. Second, doctors could become more experienced in treating the disease. Third, high death rate in the initial time period could be caused by a shortage of medical equipment supply. Last but not least, people voluntarily adjust their behaviors to reduce the transmission of COVID even without any government policies. I doubt whether the methodology taken by the authors can tease out the impact of voluntary adjustments and get a clean estimate of the policy effects. Thus, the decrease in mortality is caused by multiple factors, including government intervention policies. Attributing all decrease in mortality to intervention policies will overestimate the benefits of these policies.

Costs:

The authors estimate the cost of intervention policies by calculating the potential number of deaths due to the economic downturn caused by the interventions. The key parameter in this calculation is the cost-to-death ratio. However, the authors do not make a clear distinction between short run income shocks and the differences in income levels. Theoretically, the impacts of income shocks can be very different from income levels. If people expect that the income level will return to normal, they may not adjust their behavior. Many estimates cited by the authors estimates the cost-to-death ratio using the differences in income levels. It is questionable whether these estimates can be applied to the COVID-19 pandemic case, which is more appropriately described as an income shock.

Reviewer #2: This study is important because it reveals the impact of the COVID-19 pandemic in the USA in the first six months. The impact of the COVID-19 pandemic has been demonstrated by cost-benefit analysis. The most troubling aspects of the paper are presented under the following heads.

1. On page 5, lines 97-98 indicate that non-Covid-19 deaths are considered as costs. The reason for this has been shown as economic downturn. However, the cost of covid-19 should cover both the losses in the gross domestic product and the health expenses associated with Covid-19. On the other hand, the authors mentioned a different methodology when measuring the cost of Covid-19 under the analysis subheading in the method section (lines 130-149). The explanations in the analysis section are suitable as methods. Therefore, the sentence on lines 97-98 needs to be revised.

2. There are some writing errors in this study (eg Page 6, line 127, suc as COVID-19l). The writing errors should be carefully read and corrected by the authors.

3. I think the conclusion part of this study is not enough. The authors state that the net effect of Covid-19 on quality-adjusted life expectancy is ambiguous. It is suggested that the author explain a little more about this. The authors are expected to evaluate the possible effects of Covid-19 on quality-adjusted life expectancy.

6. PLOS authors have the option to publish the peer review history of their article (what does this mean?). If published, this will include your full peer review and any attached files.

Reviewer #1: No

Reviewer #2: No

---

## [Author Response · Author response to Decision Letter 0]

21 Oct 2021

Reviewer #1: 

Report for “Lives Saved and Lost in the First Six Month of the US COVID-19 Pandemic: A Retrospective Cost-Benefit Analysis”

This article evaluates the benefits and costs of U.S. COVID-19 mitigation policy interventions. In terms of benefits, the authors calculate the number of lives saved by subtracting the observed actual number of deaths from the unmitigated potential number of deaths. In terms of costs, the authors estimate the potential number of deaths due to the economic downturn caused by the intervention policies, which is calculated by multiplying the GDP loss in this period with the cost-to-death ratio taken from economic literature.

I have major concerns with respect to the methods used to estimate both the benefits and costs of the intervention policies.

Benefits:

The authors estimate the benefits of intervention policies by subtracting the actual number of deaths from the unmitigated potential number of deaths. By doing so, the authors assume that the decrease in COVID mortality in this time period is all caused by government mitigation policies, which is a questionable assumption in my opinion. First, the virus itself could become less deadly over time. Second, doctors could become more experienced in treating the disease. Third, high death rate in the initial time period could be caused by a shortage of medical equipment supply.

Response: Thank you for the comment. We use a range of infection fatality ratios between 0.53 and 0.82, based on a systematic review of studies that used data from the first 6 months of the pandemic. [1] In updating our literature search for this revision, we included a newly published systematic review of age-specific IFRs that produced evidence consistent with the overall IFR in the 0.53-0.82 IFR range.[2] The 0.53-0.82 range of IFR estimates is based on observed mortality during that time, therefore it incorporates epidemiological (e.g., virus becoming less deadly) and healthcare (e.g., doctors getting more experience in treating COVID, medical supplies becoming more readily available) trends mentioned by the reviewer. Although a lower median IFR of 0.23 was reported by the WHO[3] across 51 locations globally, the same study reported that among the 11 US states included in the report, the IFR ranged from 0.08 in Utah and 0.20 in Missouri, to 1.54 in Connecticut and 1.63 in parts of Louisiana. Overall, locations that had higher number of deaths (500 of more cases per million population), like most of the US, had a median IFR of 0.57. [3] Therefore, we believe that the IFR range of 0.53-0.82 reflects the current best understanding of COVID-19 mortality during the first 6 months of the pandemic. 

As context for the magnitude of the IFR estimates used in our study, it may be helpful to note that our IFR range is considerably lower than the one used in the early estimates originally reported by the WHO and CDC in March 2020 based on initial data from the Wuhan province in China (between 1.2 and 3.4) which at the time gave rise to early predictions of the potential unmitigated death toll between 2.2 and 2.9 million lives. [4, 5, 6]

We expanded on these points on pp 9, and 13-14 of the revised manuscript (page numbers here and henceforward are based on the tracked version).

Last but not least, people voluntarily adjust their behaviors to reduce the transmission of COVID even without any government policies. I doubt whether the methodology taken by the authors can tease out the impact of voluntary adjustments and get a clean estimate of the policy effects. Thus, the decrease in mortality is caused by multiple factors, including government intervention policies. Attributing all decrease in mortality to intervention policies will overestimate the benefits of these policies.

Response: The reviewer is correct in pointing out that we attribute all reduction in COVID-19 mortality (relative to the unmitigated counterfactual) to the public health measures (lockdowns, social distancing recommendations, masking recommendations), even though some voluntary behavioral modifications would likely have taken place among the public—even in the absence of these government interventions. It is not possible to empirically isolate self-induced voluntary behavioral changes that the public would have initiated without any government policies from behavioral changes induced by government regulations through either mandates or public awareness campaigns. Inability to quantify the self-initiated (not induced by government intervention) component of behavioral modification means that our analysis may overstate the effect of the government intervention on COVID-19 lives saved. However, self-induced voluntary restrictions on activity also impacted the economy in the same way as government-imposed/induced restrictions. Because behavioral adaptations impact both sides of the cost-benefit equation (lives saved and lives lost to economic shutdown), it is unclear how this may have affected our findings.

We expanded our discussion on pp 14-15 of the revised manuscript.

Costs:

The authors estimate the cost of intervention policies by calculating the potential number of deaths due to the economic downturn caused by the interventions. The key parameter in this calculation is the cost-to-death ratio. However, the authors do not make a clear distinction between short run income shocks and the differences in income levels. Theoretically, the impacts of income shocks can be very different from income levels. If people expect that the income level will return to normal, they may not adjust their behavior. Many estimates cited by the authors estimates the cost-to-death ratio using the differences in income levels. It is questionable whether these estimates can be applied to the COVID-19 pandemic case, which is more appropriately described as an income shock.

Response: We agree with the reviewer. A salient limitation of the regulation-attributable mortality literature is that cost-to-death estimates are derived from adjusted cross-sectional comparisons of mortality across individuals with different observed incomes, not clearly distinguishing between long-term income differences and short-term income variability. Theoretically, a short term loss of income, like during the economic recession that followed the COVID-19 lockdowns in the US, may impact individuals less (or differently), compared to a persistent economic disadvantage over the course of a lifetime. (67, 68) Further, the negative impact of the economic downturn on personal incomes was partly ameliorated by the US government’ COVID-19 economic relief spending providing assistance to workers, families, and businesses. Therefore, our study may have overestimated mortality attributable to government regulation and overstated its cost. 

We expanded our discussion on pp 15-16 of the revised manuscript.

Reviewer #2: 

This study is important because it reveals the impact of the COVID-19 pandemic in the USA in the first six months. The impact of the COVID-19 pandemic has been demonstrated by cost-benefit analysis. The most troubling aspects of the paper are presented under the following heads.

1. On page 5, lines 97-98 indicate that non-Covid-19 deaths are considered as costs. The reason for this has been shown as economic downturn. However, the cost of covid-19 should cover both the losses in the gross domestic product and the health expenses associated with Covid-19. On the other hand, the authors mentioned a different methodology when measuring the cost of Covid-19 under the analysis subheading in the method section (lines 130-149). The explanations in the analysis section are suitable as methods. Therefore, the sentence on lines 97-98 needs to be revised.

Response: We thank the reviewer for pointing out a lack of clarity in our explanation. Our goal is to evaluate the benefit and cost, of the public health intervention (lockdowns, masking mandates, etc)—in terms of COVID-19 lives saved versus lives potentially lost as the result of the economic downturn. In public health accounting (the societal approach we take in the manuscript), medical treatments represent economic activity (labor, medical supplies, etc) and are accounted for as part of the gross national product accounting on the cost side of the equation. All medical expenditures that occurred during the pandemic are being reflected at the national level. We agree that the sentence on lines 97-98 was poorly written. It is now revised: “The cost was the economic downturn (loss of national income during the first 6 months of the pandemic) and its attributable mortality.”

Please see p 5 (here and henceforward, page numbers reference the tracked version). 

2. There are some writing errors in this study (eg Page 6, line 127, suc as COVID-19l). The writing errors should be carefully read and corrected by the authors.

Response: We corrected the typographical error on page 6 and had the manuscript professionally copyedited. These changes are highlighted throughout the manuscript.

3. I think the conclusion part of this study is not enough. The authors state that the net effect of Covid-19 on quality-adjusted life expectancy is ambiguous. It is suggested that the author explain a little more about this. The authors are expected to evaluate the possible effects of Covid-19 on quality-adjusted life expectancy.

Response: Thank you for the comment. We added more context to the first paragraph of the discussion were we report on our findings for lives saved and lost, versus life expectancy gained and lost, as a result of the intervention. We also clarified the sentence in the conclusion, referring to the additional text now included in the discussion.

Please see pp 11-12 and p 17.

References

1. Meyerowitz-Katz G, Merone L. A systematic review and meta-analysis of published research data on COVID-19 infection-fatality rates. Int J Infect Dis. 2020;S1201(20):32180–9.

2. Levin AT, Hanage WP, Owusu-Boaitey N, Cochran KB, Walsh SP, Meyerowitz-Katz G. Assessing the age specificity of infection fatality rates for COVID-19: systematic review, meta-analysis, and public policy implications. Eur J Epidemiol. 2020;35(12):1123-38.

3. Ioannidis J. Infection fatality rate of COVID-19 inferred from seroprevalence data. Bulletin of the World Health Organization. 2021;99(1):19-33F.

4. COVID-19 Pandemic Planning Scenarios: Center for Disease Control and Prevention; 2020 [Available from: https://web.archive.org/web/20200522214936/https://www.cdc.gov/coronavirus/2019-ncov/hcp/planning-scenarios-h.pdf.

5. Ferguson NM, Laydon D, Nedjati-Gilani G, Natsuko Imai, Ainslie K, Baguelin M, et al. Report 9: Impact of non-pharmaceutical interventions (NPIs) to reduce COVID-19 mortality and healthcare demand: Imperial College COVID-19 Response Team; Mar 16 2020 [Available from: https://www.imperial.ac.uk/media/imperial-college/medicine/sph/ide/gida-fellowships/Imperial-College-COVID19-NPI-modelling-16-03-2020.pdf.

6. Walker PG, Whittake C, Watson O, Baguelin M, Ainslie KEC, Bhatia S. Report 12 - The global impact of COVID-19 and strategies for mitigation and suppression: Imperial College London; March 26 2020 [Available from: https://www.imperial.ac.uk/mrc-global-infectious-disease-analysis/covid-19/report-12-global-impact-covid-19/.

---

## [Decision Letter · Decision Letter 1]

10 Dec 2021

Lives Saved and Lost in the First Six Month of the US COVID-19 Pandemic: A Retrospective Cost-Benefit Analysis

PONE-D-21-09447R1

Dear Dr. Olga Yakusheva,

We’re pleased to inform you that your manuscript has been judged scientifically suitable for publication and will be formally accepted for publication once it meets all outstanding technical requirements.

Kind regards,

Carlos Alberto Zúniga-González, Ph.D

Academic Editor

PLOS ONE

Additional Editor Comments (optional):

Dear authors, I have made the decision to accept the manuscript, I consider that the point of the methods used to estimate benefits and costs related to intervention policies is debatable and I think the approach you give it to measure the mortality rate in the first 6 months of a pandemic; It has been a well-discussed manuscript, so I congratulate you for your contribution and effort in this type of research.

Reviewers' comments:

Reviewer's Responses to Questions

**Comments to the Author**

1. If the authors have adequately addressed your comments raised in a previous round of review and you feel that this manuscript is now acceptable for publication, you may indicate that here to bypass the “Comments to the Author” section, enter your conflict of interest statement in the “Confidential to Editor” section, and submit your "Accept" recommendation.

Reviewer #1: (No Response)

Reviewer #2: All comments have been addressed

2. Is the manuscript technically sound, and do the data support the conclusions?

Reviewer #1: No

Reviewer #2: Partly

3. Has the statistical analysis been performed appropriately and rigorously? 

Reviewer #1: Yes

Reviewer #2: N/A

4. Have the authors made all data underlying the findings in their manuscript fully available?

Reviewer #1: Yes

Reviewer #2: Yes

5. Is the manuscript presented in an intelligible fashion and written in standard English?

Reviewer #1: Yes

Reviewer #2: Yes

6. Review Comments to the Author

Reviewer #1: I still have my previous concern that the estimation of the decrease in mortality due to intervention policy is not “causal”. The decrease in COVID mortality in the lockdown period is not all caused by intervention policies. The paper lacks a proper methodology to identify the causal effect, for example comparing a state with strict lockdown v.s. another state with less strict lockdown, which is a quite standard methodology in the discipline of economics. Without a causal effect, the value of the cost and benefit analysis will be significantly limited.

Reviewer #2: (No Response)

7. PLOS authors have the option to publish the peer review history of their article (what does this mean?). If published, this will include your full peer review and any attached files.

Reviewer #1: No

Reviewer #2: No

---

## [Editor Report · Acceptance letter]

17 Dec 2021

PONE-D-21-09447R1 

Lives Saved and Lost in the First Six Month of the US COVID-19 Pandemic: A Retrospective Cost-Benefit Analysis 

Dear Dr. Yakusheva:

I'm pleased to inform you that your manuscript has been deemed suitable for publication in PLOS ONE. Congratulations! Your manuscript is now with our production department. 

Kind regards, 

on behalf of

Dr. Prof. Carlos Alberto Zúniga-González 

Academic Editor

PLOS ONE